# Vitamin C Maintenance against Cell Growth Arrest and Reactive Oxygen Species Accumulation in the Presence of Redox Molecular Chaperone *hslO* Gene

**DOI:** 10.3390/ijms232112786

**Published:** 2022-10-24

**Authors:** Akihiro Kaidow, Noriko Ishii, Shingo Suzuki, Takashi Shiina, Hirokazu Kasahara

**Affiliations:** 1Department of Biology, School of Biological Sciences, Tokai University, Sapporo 005-8601, Japan; 2Hokkaido Regional Research Center, Tokai University, Sapporo 005-8601, Japan; 3Department of Molecular Life Science, School of Medicine, Tokai University, Isehara 259-1193, Japan

**Keywords:** vitamin C, redox chaperone, reactive oxygen species metabolism, *hslO*

## Abstract

Chromosome damage combined with defective recombinase activity renders cells inviable, owing to deficient double-strand break repair. Despite this, *recA polA* cells grow well under either DNA damage response (SOS) conditions or catalase medium supplementation. Catalase treatments reduce intracellular reactive oxygen species (ROS) levels, suggesting that *recA polA* cells are susceptible to not only chronic chromosome damage but also ROS. In this study, we used a reducing agent, vitamin C, to confirm whether cell growth could be improved. Vitamin C reduced ROS levels and rescued colony formation in *recAts polA* cells under restrictive temperatures in the presence of *hslO*, the gene encoding a redox molecular chaperone. Subsequently, we investigated the role of *hslO* in the cell growth failure of *recAts polA* cells. The effects of vitamin C were observed in *hslO^+^* cells; simultaneously, cells converged along several ploidies likely through a completion of replication, with the addition of vitamin C at restrictive temperatures. These results suggest that HslO could manage oxidative stress to an acceptable level, allowing for cell division as well as rescuing cell growth. Overall, ROS may regulate several processes, from damage response to cell division. Our results provide a basis for understanding the unsolved regulatory interplay of cellular processes.

## 1. Introduction

In combination with recombinase defects, chromosome breakage and double-strand break repair deficiencies render cells inviable. However, in *Escherichia coli recAts polA* cells, viability was retained when DNA damage response was evoked by constitutive expression of a *lexA* (Def) mutation [1,2]. We had previously shown that *recA polA* cell growth arrest was partially due to the accumulation of reactive oxygen species (ROS) and hydrogen peroxide under restrictive temperatures and rich medium conditions [2,3].

*E. coli* frequently encounter oxygen radicals produced as metabolic byproducts. These oxygen radicals are metabolized into hydrogen peroxide, which, in turn, is detoxified into water by catalase. Oxygen radicals also serve as a stress marker and as an effector owing to their high reactivity. Oxidative damage by superoxide anions directly suppresses metabolic activities that depend on enzymes that conjugate with iron and flavin. Oxygen radical accumulation causes stress and cell death following lethal treatments [4]. Yet, controversy remains as to whether the accumulation of oxygen radicals causes cell death or bacteriostasis [5]. Under thymine deprivation, single-strand regions and cellular oxygen radicals accumulate [6]. Furthermore, Hong et al. reported that lethal effects on the cells resulted from the accelerated accumulation of oxygen radicals in a self-replicating manner, overwhelming primary damage reparation [4]. Vitamin C plays an important role in maintaining proper oxidation–reduction balance through the neutralization of ROS and reactive nitrogen species (RNS) formed during cellular metabolism [7]. The ascorbate–glutathione cycle is involved in the aforementioned process; however, it is unknown whether this cycle is also functional in *E. coli* cells.

Redox signaling functions through redox chaperones and transcription factors, which use sensor regions to bind or react with molecules such as ROS and hydrogen peroxide. These proteins regulate the transcription or enzymatic activity of other molecules through structural changes caused by their respective oxidation or reduction [8,9,10,11]. Redox molecular chaperones include HslO [12] and AhpC [13]. Interestingly, AhpC and vitamin C are associated at an expression level [14]; they regulate other molecules and control biological reactions via molecular state adjustments, oligomeric aggregation, and dissociation. Regarding transcription factors, OxyR is oxidized and transcriptionally activated by hydrogen peroxide [15], while *soxRS* is activated via ROS-induced intracellular state changes [16,17].

The HslO protein, encoded by *hslO*, is the redox molecular chaperone Hsp33. HslO protects organisms from oxidative stress that leads to protein unfolding. The loss of HslO function also renders cells susceptible to hydrogen peroxide [12]. HslO activation is triggered by the oxidative unfolding of the HslO redox sensor domain [18]. Further, HslO is considered a member of a recently discovered class of chaperones that requires partial unfolding for full chaperone activity [19]. The molecular basis for regulating HslO function begins with the release of prebound Zn from the well-conserved Zn-binding motif Cys-Xn-Cys-X-Cys upon oxidation and activation, causing remarkable structural changes [20,21]. Once activated, HslO protects proteins from coagulating into toxic proteins, thus preventing cellular deaths. Using the unfolding region, HslO binds the unfolding client proteins [20]. The mechanism by which the *hslO* docking surface captures the coagulated body of client proteins has been elucidated [22].

The lowering of intracellular ROS levels promotes the viability of *recA polA* cells. Additionally, the complete absence of RecA function is tolerated in *polA* cells, indicating that *recA polA* cell mortality or growth arrest could be related to intracellular ROS levels rather than a deficiency in DSB repair. Furthermore, redox chaperone *hslO* is required for *recA polA* cells to survive [2]. However, the relationship between DNA damage, ROS production, and cell proliferation remains unclear. In this study, we explore the growth of *recAts polA* cells in the presence of a reducing reagent, vitamin C. We further elucidate a potential interplay between *hslO* in this suppressive process of *recAts polA* lethality and the coordination of cellular processes. We also aimed to confirm whether ROS are an important factor in explaining the sequence of events from DNA damage to growth arrest (or lethality) observed in *recAts polA* cells. We found that *hslO*^+^ cells were more proficient in the suppression of lethality after treatment with vitamin C than *ΔhslO* counterparts. This serves as a useful key in understanding what occurred in those cells. Our findings regarding ROS, a potential inhibitor for cell growth, will generate new aspects of redox signaling to explore in *E.*
*coli*.

## 2. Results

### 2.1. recAts polA Cells Form Colonies in Restrictive Temperatures in the Presence of Vitamin C

In a previous report, we demonstrated that *recA polA* cells were capable of colony formation in catalase-supplemented media either at a restrictive temperature (42 °C) [2] or in a rich medium [3]. Since catalase is a hydrogen peroxide-degrading enzyme, this suggested that the inability of *recAts polA* cells to grow at restrictive temperatures resulted from hydrogen peroxide accumulation. Vitamin C is an antioxidant that scavenges superoxide, hydrogen peroxide, and singlet oxygen in vitro [23]. This prompts the question of whether vitamin C affects cellular responses to hydrogen peroxide metabolism (Figure 1). In the spot assay, *E. coli* cell particles were spotted on plates and then incubated at either permissible (30 °C) or restrictive temperatures (42 °C). As seen in Figure 1, AB1157 derivative cells, including TK3077 (*recAts polA*), TK3276 (*recAts polA ΔhslO*), TK3473 (*recAts polA ΔhslO p*EXvec), and TK3474 (*recAts polA ΔhslO p*EX*hslO*), exhibited similar growth at 30 °C. In contrast, cells failed to grow at 42 °C without vitamin C, except for TK3077 cells spotted with 2 × 10^6^ particles, which were confirmed to be temperature-sensitive. Interestingly, the *recAts polA* strain (TK3077) could grow on plates at 42 °C when plates were supplemented with either 100 μM or 1 mM, but 10 μM vitamin C was insufficient to encourage growth. A 100- to 1000-fold increase in colony-forming ability with 100 μM vitamin C was observed when compared with cells without vitamin C at 42 °C. In contrast to the TK3077 cells, the *recAts polA ΔhslO* strain (TK3276) failed to grow with the addition of 100 μM vitamin C at 42 °C. In addition, TK3276 cells were transformed with either vector (TK3473) or with the *hslO*-expressing plasmid *p*EX*hslO* (TK3474). As a result, TK3474 cells could form colonies at 42 °C when 2 × 10^6^ bacterial particles were spotted in the presence of 100 μM vitamin C, but not for 2 × 10^5^ particles. TK3473 cells at 42 °C failed to form colonies, even at 2 × 10^6^ bacterial particles. Therefore, it is assumed that the colony formation ability of TK3474 cells was improved more than 10-fold compared to that of TK3473 cells. Thus, vitamin C, a ROS scavenger, restored colony formation at restrictive temperatures in *recAts polA* cells.

To directly assess the effect of vitamin C on cell growth, we used liquid media, in which cell growth could be easily measured. The bacterial culture was divided into four equal volumes in an early logarithmic phase (O.D._600_ = 0.1) at 30 °C. Figure 2 shows the growth of cells under four different conditions: (a) absorbance or (b) the number of *E. coli* cells. As shown in Figure 2a, the *recAts polA* cells (TK3077) at 42 °C showed slightly improved growth at 16 h with the addition of 100 μM vitamin C (blue), compared with cells without vitamin C (red). The number of bacterial cells at 16 h is also shown in Figure 2b. However, no remarkable differences were observed regardless of vitamin C presence in TK3077 (Figure 2b). Nonetheless, a significant difference in cell particles was observed between TK3077 (in the presence of *hslO*) and TK3276 (in the absence of *hslO)* at 42 °C with the addition of 100 μM vitamin C (Welch’s *t*-test, *p* < 0.05). This indicates that *hslO* is necessary for vitamin C to stimulate a slight increment in cell growth and division. Colony formations were remarkably restored in TK3077 cells with vitamin C at 42 °C (Figure 1), in spite of only a slight increase in cell particles. Thus, it was necessary to examine whether these cell particles were membrane-damaged. Therefore, we assayed dead cells with compromised membrane integrity using propidium iodide (PI) staining. Membrane-damaged dead cells were not detected at 30 °C (Appendix A) but were detected at 42 °C, as indicated by blue arrowheads (Appendix A). However, contrary to our expectation, the membrane-damaged cell population (blue arrowheads) did not show a remarkable decrease following vitamin C addition. Meanwhile, a minor cell population, staining poorly for both PI and SYTO9 (Appendix A, red arrowheads), had shrunk. This suggests that vitamin C addition did not remarkably reduce the number of membrane-damaged cells. It is unclear whether this small change suffices to explain the observed increase in colony formation in Figure 1. Furthermore, this does not take into consideration any possible decrease in the number of anucleate cells. Therefore, we considered whether qualitative rather than quantitative changes in chromosomal DNA might be involved in the cell viability decrement at restrictive temperature in *recAts polA* cells.

### 2.2. Vitamin C Is Involved in Maintaining the Colony-Forming Ability of recAts polA Cells at Restrictive Temperatures

As shown in Figure 1 and Figure 2, *recAts polA* cells supplemented with vitamin C were able to form colonies at 42 °C and showed slightly improved growth over those without vitamin C in a liquid medium. These results might suggest that the effect of vitamin C did not support cell growth directly, but rather indirectly promoted colony formation. Therefore, we attempted to compare the effect of vitamin C on viability, a major qualitative change that directly relates to cell growth, in both TK3077 and TK3276. We examined the effects of preincubation treatments with vitamin C at either permissive (30 °C) or restrictive temperatures (42 °C) before cells were spotted on plates. Following the treatment, the survival of cells was determined on plates at either 30 °C (Figure 3a,b) or 42 °C (Figure 3c,d). With regard to the plate culture at 30 °C, the *recAts polA* cells with *hslO*^+^ (TK3077) did not show any response to the addition of vitamin C with 30 °C permissive pretreatment (lanes 1 and 2). TK3077 cells decreased in colony formation in restrictive pretreatment, agreeing with the temperature-sensitive phenotype of TK3077 cells (lanes 3). Meanwhile, cells with 100 μM vitamin C supplementation and 42 °C restrictive pretreatment maintained the ability to form colonies, >10-fold better than those without vitamin C supplementation (lanes 3 and 4). Interestingly, colony-forming abilities with vitamin C for 42 °C restrictive pretreatment remained 10-fold better in comparison to those of 30 °C permissive pretreatment (lanes 2 and 4). These results indicated that TK3077 cells with 42 °C restrictive pretreatment maintained their cell integrity until colony formation, owing to the addition of vitamin C. We further examined treatment effects on TK3276 cells (the *hslO* deletion derivative). These cells showed no substantial difference in colony-forming abilities between those with and without vitamin C supplementation at both 30 °C and 42 °C treatment (lanes 5–8). Interestingly, TK3276 cells with 30 °C treatment showed better colony formation than TK3077 cells (lanes 1 and 5). Similar colony-forming abilities were observed between the *hslO*^+^ and *hslO* deletion mutant cells after 42 °C treatment without vitamin C (lanes 3 and 7). Remarkably, TK3077 cells formed colonies 100 times better than TK3276 cells with 42 °C treatment and vitamin C (lanes 4 and 8). Regarding the plate culture at 42 °C, TK3077 cells with vitamin C formed colonies mainly at 2 × 10^6^ (lanes 11 and 12). These results suggest retention of colony-forming capabilities with vitamin C in 42 °C treatment, which would not otherwise have occurred from acquiring a temperature-resistant mutation. Therefore, vitamin C was somehow maintaining cellular integrity, which likely resulted in the observed colony formation. Furthermore, *hslO* contributed to whether cells could grow or not in the presence of vitamin C after restrictive pretreatments.

### 2.3. Vitamin C Reduced Accumulation of Intracellular ROS Levels at Restrictive Temperatures in hslO^+^ recAts polA Cells

We investigated the effect of vitamin C on the accumulation of intracellular ROS levels in *recAts polA* cells at a restrictive temperature (Figure 4). TK3077 (*recAts polA*) and TK3276 (*recAts polA ΔhslO*) cells were cultured at 30 °C and divided into four equal parts in the early logarithmic growth phase (O.D._600_ = 0.1). Histograms showed the accumulation of intracellular ROS levels at 16 h after cultivation at either 30 °C or 42 °C with or without 100 μM vitamin C supplementation. The accumulated intracellular ROS levels in TK3077 cells were almost the same under the permissive condition (30 °C) with or without vitamin C supplementation (Figure 4a). In contrast, at the restrictive temperature (42 °C), TK3077 cells with vitamin C (blue) had lower intracellular ROS levels compared with those without vitamin C (red). Meanwhile, the *hslO* deletion derivative cells (TK3276) at 30 °C had intracellular ROS accumulation levels similar to those of TK3077 cells at 30 °C, with and without vitamin C supplementation (Figure 4b). However, the intracellular ROS levels of TK3276 increased even with the addition of vitamin C at 42 °C. This was in contrast to TK3077 cells, in which intracellular ROS levels decreased with the addition of vitamin C.

Next, we analyzed the dose-dependent response in mean ROS levels towards vitamin C. We observed poor dose response; however, the effect of vitamin C was observed at 100 μM in TK3077 (Appendix A). This poor dose response and unstable results at both 300 and 1000 μM might be due to a hormesis effect [24], in which a response is not always according to the dosage.

We further statistically evaluated the effect of 100 μM vitamin C on the intracellular ROS levels at 30 °C and 42 °C (Figure 5). As shown in Figure 5a, mean intracellular ROS levels were not influenced by vitamin C in TK3077 and TK3276 cells at 30 °C. However, those of TK3077 cells at 42 °C were significantly reduced with the addition of 100 μM vitamin C (Welch’s *t*-test; *p* < 0.05, n = 7). Conversely, mean intracellular ROS levels increased upon the addition of vitamin C in the TK3276 cells. Thus, the vitamin C effect on intracellular ROS levels differs between TK3077 and TK3276 cells.

Further, inoculation experiments were also performed to confirm the results above at either 30 °C or 42 °C, in the absence or presence of vitamin C. This method was especially convenient for measuring intracellular ROS levels (Figure 5b). The addition of vitamin C significantly decreased mean ROS levels in TK3077 cells (Figure 5b) (*p* < 0.05, n = 7). Contrary to the results in Figure 5a, ROS levels in TK3276 cells responded to vitamin C in this inoculation experiment. This might have resulted from the difference in the growth phase of the two experimental systems: the system in Figure 5a used the logarithmic growth phase, while the system in Figure 5b used the stationary phase. Growth in Figure 5a,b corresponded to that shown in Figure 2a,b and Appendix A, respectively. In the inoculation experiment, unlike experiments divided at the logarithmic phase, neither TK3077 nor TK3276 cells showed any growth at 42 °C (Appendix A). These growth differences may have resulted from the average intracellular ROS levels in the two experimental systems. This hypothesis is studied below, at least with regard to cell growth and ROS levels.

### 2.4. Vitamin C Affected Intracellular ROS Levels in recAts polA Cells by Synchronizing ROS Levels with Growth

To interrogate the relationship between ROS levels and growth, we used results from Figure 5a to construct a scatterplot in two dimensions with growth on the vertical axis and mean ROS levels on the horizontal axis (Figure 6). When vitamin C was added at 42 °C (blue), the mean intracellular ROS levels were reduced in TK3077 cells (Figure 6a) compared with those in TK3276 cells (Figure 6b). Measurement points for TK3276 cells showed neither arbitrary units in growth at around 200–600 RFU (relative fluorescence unit, RFU) nor mean ROS levels at around 175–200 RFU. This suggests that TK3276 cells failed to alleviate ROS levels with vitamin C due to the absence of *hslO,* resulting in a lack of proficient growth recovery. Thus, vitamin C countered the growth inhibition of TK3077 cells at 42 °C, but not that of TK3276 cells.

We wondered whether *recAts polA* cells synchronized in growth and ROS levels after 16 h of incubation (Figure 6a,b). We further confirmed this possibility with cell counts in culture and ROS levels. As shown in Figure 6c, both TK3077 and TK3276 cells cultured at 30 °C were aligned at ROS levels at around 175 RFU. Meanwhile, TK3276 cells cultured at 42 °C were aligned at 2.5 × 10^8^ cells/mL, corresponding to nearly 2.5 times as much as an initial inoculated cell concentration. Contrary to these results, TK3077 cells incubated at 42 °C without vitamin C had aligned in both ROS levels and the cell concentration, resulting in an L-figured distribution. Furthermore, TK3077 cells incubated at 42 °C with vitamin C were mainly aligned in ROS levels at 175 RFU. In summary, vitamin C, in the presence of *hslO*, would partly improve both growth and mean ROS levels at 42 °C. Further, these results suggest that there is a growth threshold in *recAts polA* cells with regard to mean intracellular ROS levels after 16 h. Thus, these results indicated that vitamin C proficiently restored cell growth in the presence of *hslO* among *recAts polA* cells. Interestingly, it is possible that cell division processes might have caused *recAts polA* cells’ growth failure at restrictive temperature as they divide only once. Cell division could be inhibited through multiple processes, including DNA replication and segregation, since it is the final stage of quality control. Thus, DNA levels were of interest in *recAts polA* cells.

### 2.5. Vitamin C Influences a Convergence of Cell Population with Chromosomal DNA Level Rather Than the Temperature-Sensitive Recombination of the recAts Mutation

As previously mentioned, *recAts polA* cells might postpone (or inhibit) cellular processes, including cell division, due to DNA damage. It was likely that DNA metabolism including a completion of DNA replication was a prerequisite for chromosome segregation and so on. The initiation of DNA replication is tightly associated with these cell masses and growth rates [25]. Thus, slowly growing cells and cells in the stationary phase possess chromosomal DNA levels with one or two chromosomes. Accordingly, we focused on monitoring the chromosome status of *recAts polA* cells with their DNA levels through this cell cycle progression. The addition of vitamin C was involved in the maintenance or promotion of colony formation in *recAts polA* cells at restrictive temperatures. Recalling Appendix A, most *recAts polA* cells did not show notable membrane damage at restrictive temperatures; simultaneously, vitamin C had no effect on damaged populations. Cell death or growth arrest is unlikely to occur suddenly without any precursory phenomenon. Therefore, as a prerequisite for colony formation, we compared the effect of vitamin C on chromosome status through quantitative chromosome staining with PicoGreen; simultaneous deficiencies in *recA* and *polA* are thought to influence chromosome integrity. To compare the chromosome status, ploidy analysis using flow cytometry was performed. Cells incubated at 30 °C showed three bands corresponding to DNA quantity, as in one, two, and three ploidies (Figure 7). Cells incubated at 42 °C did not show these bands. Nonetheless, TK3077 cells supplemented with vitamin C at 42 °C still showed major converging bands with approximately two ploidies. The bands appearing from TK3077 cells with vitamin C supplementation at 42 °C were similar to those appearing at 30 °C. These results were also analyzed with histograms (Appendix A). The histograms show sharp peaks corresponding to converged chromosomes at 30 °C; however, those at 42 °C are difficult to distinguish. As described above, even in randomly growing cells, the majority of cells converged with several DNA levels owing to both the completion of DNA replication and the relatively short replicating time, thus presenting converged histograms or bands. Therefore, DNA levels in these experiments were converged in one or two ploidies. Interestingly, histograms in TK3276 cells show that DNA quantity had increased at 42 °C compared with that at 30 °C (Appendix A, lower panels). TK3077 cells with vitamin C supplementation at 42 °C showed several peaks corresponding to converged DNA content (Appendix A, upper right). In agreement with the results presented in Appendix A, the anucleate cell population was slightly decreased upon the addition of vitamin C in TK3077 cells but not in TK3276 cells (thick green arrowhead in Figure 7 and Appendix A). It is unclear whether this decrease in anucleate cells affects growth failure. We also quantified DNA content as a function of time in the presence or absence of vitamin C (Appendix A). No remarkable differences were observed between TK3077 and TK3276 cells. Thus, vitamin C mainly alters chromosome states such as chromosome ploidy rather than the amount of DNA. Naturally, the stationary phase cells have one or two chromosomes due to chromosomal replication and segregation. Therefore, chromosome ploidy could be one of the phenotypical markers for the successful progression of cellular processes towards the stationary phase. TK3077 and TK3276 cells showed irregular ploidies even in permissive temperature (30 °C), and those abnormalities were enhanced at restrictive temperature (42 °C). This enhancement of abnormalities was partially alleviated in TK3077 cells with vitamin C, even at 42 °C. These results suggest that TK3077 cells could complete their chromosomal replication in the presence of vitamin C at 42 °C, thus supporting colony formation. Without vitamin C, TK3077 and TK3276 cells failed to complete or delayed the processes toward the stationary phase. Therefore, qualitative rather than quantitative changes in chromosomes might be involved in the increase of viability with the addition of vitamin C, through synchronization between growth and ROS levels.

It could be argued that adding vitamin C might influence the temperature sensitivity of the recombination reaction in *recAts* cells. Since *E. coli* cells with the *polA25* mutation had low P1 transduction efficiency, we used *polA*^+^ cells with *recAts* mutation to evaluate the effect of vitamin C on temperature sensitivity. *recA*^+^ parental cells (AQ10459) showed a transduction frequency of nearly 10^−5^ at both 30 °C and 42 °C, with or without 100 μM vitamin C (Figure 8). AQ10546 *recAts* cells showed a transduction frequency of about 5 × 10^−6^ with or without vitamin C at 30 °C. Likewise, at 42 °C, the transduction frequency of AQ10546 cells (at 1 × 10^−7^) did not differ between treatments with and without vitamin C. These results suggested that adding vitamin C did not influence the transduction efficiency of either *recA*^+^ or *recAts* cells. Therefore, we presume that suppression of temperature sensitivity did not result from a restoration of genetic recombination due to vitamin C. This agreed with our previous result that *ΔrecA polA* cells were viable. Thus, these results supported the hypothesis that the progression of cellular processes toward the convergence of cell population with the completion of DNA replication was affected in *recAts polA* cells.

## 3. Discussion

Our previous study on recAts polA lethality reported that intracellular ROS accumulation was associated with lethality at the restrictive temperature [2]. However, why *hslO* was required to suppress *recAts polA* lethality was unclear. Therefore, we intended to elucidate the role that *hslO* played in the suppression of lethality. In this study, we determined that the accumulation of radical oxygen was partly alleviated by the addition of vitamin C and the presence of *hslO*. *recAts polA* lethality (or growth inhibition) was ameliorated by the lowering of ROS levels in cells by *hslO*. These findings are in agreement with our previous report that *recAts polA lexA51* cells required *hslO* for suppression of temperature sensitivity. Our present investigation has suggested that *hslO*, in the presence of vitamin C, plays an important role in cellular function integrity, including the completion of chromosome replication and colony formation via lowering ROS levels. In other words, ROS production upon chromosome damage would negatively regulate normal processes of the cell (e.g., through redox signaling and prolonged damage). This leads to a gradual loss of cell functional integrity and, eventually, cell death. In this case, the reduction in ROS alone might be insufficient to accomplish the progression of cell growth. These possibilities are further discussed later.

Even though hydrogen peroxide and superoxide anions do not directly oxidize DNA, these molecules contribute directly or indirectly to the production of highly reactive hydroxy radicals that damage bacterial chromosomes [26]. Thus, reducing the concentration of oxygen radicals early can prevent or delay damage. Due to the electron-donating ability of vitamin C, it could act as a free radical scavenger and reduce iron in its highly oxidized state to Fe^2+^ [23]. The effects of vitamin C are still controversial, although many trials are underway. This might be due to hormesis-like effects [24]: Kontek et al. reported that vitamin C ameliorated DNA damage only in the presence of H_2_O_2_ [27]. This indicates that the addition of reducing reagents could regulate the excess oxidative intracellular state, which is consistent with our results. Thus, these results suggest that *E. coli* can mediate the damage caused by ROS with the addition of vitamin C.

Furthermore, vitamin C could reduce H_2_O_2_, ROS, and disulfide bonds through cellular metabolism [28]. *E. coli* possesses a weak dehydroascorbate reductase activity [29], but it is unknown whether the ascorbate–glutathione cycle is also functional in *E. coli* cells. However, the bacterial peroxiredoxin AhpC might contribute to the cycle via the deglutathionylation cycle [13], and AhpC and vitamin C are associated at the expression level [14], suggesting a response to oxidative stress generated during the aerobic metabolism of vitamin C. Vitamin C could prevent the detrimental effects of H_2_S, which reduces glutathione levels in *E. coli* [30], and AhpC was resistant to inactivation by peroxidation [31] through suppressing aggregation of client proteins in heat shock conditions. Interestingly, AhpC was reported to function as a redox molecular chaperone, similar to HslO, to regulate the oligomer state of the target proteins. These findings suggest a possible crosstalk between vitamin C and redox molecular chaperones.

Molecular redox chaperones such as *hslO* play a crucial role under oxidizing conditions. *hslO* is unique for promptly detecting oxidation stress, a possible cause of the unfolding of proteins, and thus becomes activated. Furthermore, once activated by oxidization, *hslO* protects proteins from coagulating and forming toxic proteins, which then protects bacterial cells from cellular death. These results agree with our observations, leading to a new direction in research of both DNA damage response and *hslO*; however, direct linkages have not been fully understood yet. Using the unfolding region, the *hslO* binds with the unfolding client proteins [20]. The candidates for the client proteins that possess a DNA-repairing ability have not yet been reported, although how HslO captures client proteins on its docking surface has been elucidated [22]. It is likely that some proteins that are not clients of other redox chaperones could be associated with the Srp pathway. Protein aggregates might be involved in cellular process progression and general intracellular ROS regulation.

Notably, in our study, the abnormalities of ploidies were partially restored in TK3077 cells with vitamin C at 42 °C. Chromosome ploidies are ordinarily 2^n^, resulting from proper termination of chromosome replication, segregation, and septation in *recA*^+^ cells at the stationary phase. However, *recA*^−^ cells do not show a 2^n^ pattern in the stationary phase because of RecA function deficiencies, causing a failure of proper chromosome segregation in catenated chromosomal DNA. Even at a permissive temperature, *recA200* (*recAts*) mutation caused insufficient RecA function, which was exacerbated at restrictive temperatures. Lanzov et al. reported that the *recAts* mutation caused hyper-recombination at a restrictive temperature [32]. Kogoma reviewed interplays between recombination and replication [33]. Therefore, at the restrictive temperature, it is possible that *recAts polA* cells caused hyper-replication, and replicative stress might be elevated in *recAts polA* cells. Of note, *ΔrecA polA* cells failed to grow in a rich medium where replicative stress would be elevated [3].

We have reported in a previous paper that significant chromosome breakdown could not be observed [2]. However, we were not able to identify what causes growth failure of the *recAts polA* cells. Therefore, we instead focused on the process leading to ROS accumulation and also characterized the damaged chromosomes. As a result, *recAts polA* cells could grow in the presence of both the *hslO* gene and vitamin C under restrictive conditions. Conversely, we found that *recAts polA* cells failed in the convergence of cell population with their DNA content at the restrictive temperature; however, this phenomenon was restored in the presence of *hslO* and vitamin C. It is not yet known whether the growth failure and the convergence of cell population with DNA content are related to each other. It is interesting that slowly growing cells possess one or two ploidies, indicating that completion of replication was always observed for those slowly growing cells before cell proliferation via cell division. Meanwhile, the oxidation of DNA polymerase likely leads to their inactivation. Therefore, *hslO* and vitamin C can function together to maintain cellular oxidative conditions at allowable levels, supporting cellular metabolism and the completion of DNA replication required for the convergence of the cell population with DNA contents, thus enabling cell division and cell proliferation.

Regarding the use of a synthetic lethality experimental system for chromosome damage, we observed growth arrest of *recAts polA* cells that was ameliorated with reducing reagents such as vitamin C. Additionally, cells were growing with synchronization in growth and ROS levels even in restrictive temperatures. This is consistent with our previous observations on the effects of catalase [2,3]. The source of ROS is unclear; however, we now know how the *recAts polA* cells mediate ROS stress. Our study substantially depended on analysis with chemical probes and fluorescence dyes by flow cytometry. This approach is reliable for population analysis. Yet, it is perhaps unreliable for particular molecules and fine quantitative analysis. Notably, chromosome ploidy was restored by vitamin C at the restrictive temperature. Intracellular ROS levels could influence chromosome replication and/or segregation, indicating that these cellular processes might be the targets of ROS stress. Conclusively, these findings have opened doors to new forms of interplay between the regulation in the progression of replication and DNA damage via redox signaling. These findings will contribute to understanding how DNA damage signals are transduced into chromosome replication during cell proliferation. Regulations, including modifications of particular cellular reactions or mechanisms, might suffice for now as one thread in the larger tapestry of oxidative stress response pathways. However, another thread comes from the interplays between intracellular metabolites and individual cellular processes as well. We may be on the verge of a new era in which all the threads will come together for us to see the greater picture.

## 4. Materials and Methods

### 4.1. E. coli Strains and Media

The *E. coli* strains used in this study are represented in Appendix A and are also described as follows: TK3077 was the same as AQ10549 but restocked. AQ10459, AQ10546, AQ10865, TK3077, TK3276, TK3473, and TK3474 were AB1157 derivatives described previously. TK3019 was constructed using phage P1*vir*-mediated transduction [34,35]. Constructed with infection of P1*vir* phage from AQ10865, the Cm^r^ colony was selected. Subsequently, a temperature-resistant strain for growth was also confirmed for *lexA*^+^. The resulting cells were AQ634 *malB*::Tn*9 lexA*^+^. Cells were grown at 30 °C in M9 salts–glucose minimal (M9G) media supplemented with casamino acids (CAAs) (0.2%; Difco Laboratories, Detroit, MI, USA); thymine (1 mg/mL); thiamine (1 μg/mL); appropriate amino acids (50 μg/mL): arg, thr, leu, trp, his, pro (M9GCAA medium); and antibiotics: ampicillin (20 μg/mL), kanamycin (55 μg/mL), spectinomycin (40 μg/mL), and streptomycin (100 μg/mL).

### 4.2. Cultivation and Sampling Methods

M9GCAA liquid media (2 mL and 15 mL) were placed in test tubes and 100 mL Erlenmeyer flasks, respectively, and cultured aerobically at either 30 °C or 42 °C. Cells were inoculated with 1/100 volume of cells grown overnight on M9GCAA broth.

In shift-up experiments, cells were cultured in M9GCAA medium until O.D._600_ = 0.1. Cells were then divided into two to four equal portions for the addition of reagents. After the treatments, the cell culture absorbance values were determined at O.D._600_, and DNA content and ROS analyses were performed every 2 h for a total of 16 h. For the time-course experiment, typical sample volumes were 600 μL for O.D._600_, 200 μL for DNA content, and 4 μL for ROS analyses.

In inoculation experiments, cells without or with a predetermined menadione concentration were incubated at 120 spm for 16 h at either 30 °C or 42 °C.

### 4.3. Plasmid Construction

An in-frame *hslO* expression plasmid (*p*EX*hslO*) was constructed by introducing *Nco*I-*EcoR*V-*Nde*I site *Nde*I-blunted *p*MW119 (Nippongene, Tokyo, Japan) derivative (pTK1424) at 3 bp downstream of the Shine–Dalgano sequence of *lacZ* by PCR. Inverse primers were 5′-GATATCCATATGACCATGATTACGCCAAG and 5′-TGGATATCCATGGCTGTTTCCTGTGTGAAATTG. The inverse PCR product was digested with *EcoR*V and then self-ligated. A ligated DNA was transformed into XL-1 Blue (Agilent, California), and a blue colony was selected. *Nde*I-sensitive plasmids were further confirmed by sequencing and designated as *p*TK1434. Thus, CT sequences just before the first ATG of *lacZ* of *p*MW119 were replaced by CCATGGATATCCAT, which resulted in the addition of *Nco*I-*EcoR*V-*Nde*I sites to *lacZ* in *p*TK1434.

*hslO*/*srpC* gene fragments were amplified through 15 cycles with 94 °C/1 min, 40 °C/1 min, and 72 °C/2 min. Primers were 5′-TTAAGCTTAGCCATGGCTCATATGATTATGCCGCAACATG and 5′-TTGGATCCTGTACATTAATGAACTTGCGGATC. Amplified DNA was digested with *Hind*III and *Bam*HI and cloned into the *Hind*III and *Bam*HI sites of *p*STV28. Sequences were confirmed, and the Spc cassette was subsequently introduced into the *Sma*I site. This transient plasmid and *p*TK1434 were digested with both *Nco*I and *Kpn*I and ligated. A white colony was selected, and the resultant plasmid was designated as a *p*EX*hslO*. Spc cassette was also introduced into the *Sma*I site of *p*TK1434 as *p*EXvec.

### 4.4. Determination of Survival Fraction and Cell Recovery

For relative viability (RV) determination, after incubation in M9GCAA medium overnight at 30 °C, the cells were diluted in M9 medium without a nutrient source (M9B) and then plated on M9GCAA plates supplemented with appropriate antibiotics and incubated for 16 h at either 30 °C or 42 °C. Cell concentrations of 2 × 10^1^, 2 × 10^2^, 2 × 10^3^, 2 × 10^4^, 2 × 10^5^, and 2 × 10^6^ were spotted on plates, and the viability was determined by photographs after 16 h of cultivation at either 30 °C or 42 °C.

### 4.5. Flow Cytometry Analysis

Flow cytometry was conducted as previously described [2]. For ROS analysis using flow cytometry, staining was performed according to our previous studies [2,3]. Cell cultures (4 μL) were mixed with 12.5 μM CellRox Deep Red (16 μL) at indicated times, diluted with M9 medium without organic nutrients (M9B), and stained for 30 min at 25 °C. Stained cells (20 μL) were then diluted in M9B (200 μL). Except for cells from an agar plate, we stained cells with CellRox Deep Red alone. We then used a Becton Dickinson Accuri C6 (Becton, Dickinson and Company, Ann Arbor, MI, USA) with a 640 nm laser. First, we analyzed the cell culture for the gate derived from cell particles. We used identical side scatter signal/forward scatter signal (FSC) gates, designated as P3, and collected 50,000 events. In our experiments, the rate of events was less than 2500 events/s. To analyze the acquired data, we used the C6 software (version 1.0.264.21). Each sample was plotted as a histogram vs. the red channel (FL4-A with 675 ± 15 nm filter) or ROS content (fluorescence, channel FL4-A) as either autofluorescence by the green channel (FL-1A) or as a function of the cell size (as FSC above). DNA content analysis was carried out according to Ferullo et al. [36]. For 2-dimensional ploidy analysis, side scatter (SSC)-H intensities were measured and assigned to the *y*-axis of the dot plot. The number of particles in cultures was determined with the BD Cell Viability Kit (Becton, Dickinson and Company, 335925), following the manufacturer’s procedure.

Live–dead staining was carried out using LIVE/DEAD BacLight Bacterial Viability and Counting Kit with the manufacturer’s recommended procedure.

### 4.6. Statistical Analysis

The calculation of means was performed using Microsoft Excel 2019, and the standard error of the mean (SEM) was calculated with the *STDEV.P* function. Welch’s *t*-test was also performed using the Excel program, and statistical significance was set at *p* < 0.05.

## 5. Conclusions

Temperature sensitivity of *recAts polA* cells was suppressed with the addition of vitamin C, which is consistent with our previous observation that the addition of catalase to plate culture restored these cells. The effect of vitamin C was related to the maintenance of cell viability rather than the restoration of cell growth. This phenomenon coincides with the amelioration of ROS levels by vitamin C, resulting in the overall restoration of cellular processes and progression toward the stationary phase. These results present a possible link between ROS levels and cell viability, revealing a partial mechanism behind *recAts polA* lethality.

## Figures and Tables

**Figure 1 ijms-23-12786-f001:**
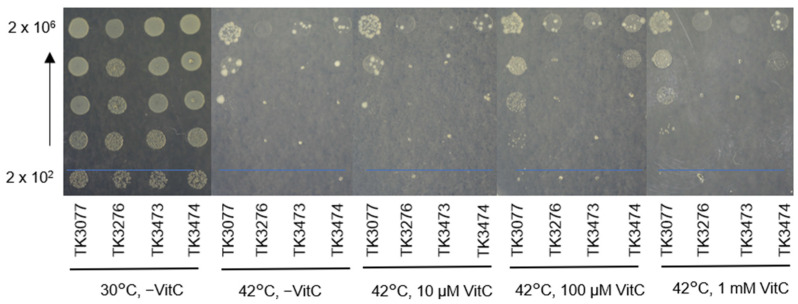
Effects of vitamin C addition to plate medium on colony formation of *recAts polA* cells. Cells were spotted with the number of particles indicated on the left-hand side and incubated at 30 °C and 42 °C for 2 days. From left to right: *recAts polA* (TK3077), *recAts polA ΔhslO* (TK3276), *recAts polA ΔhslO p*EXvec (TK3473), and *recAts polA ΔhslO p*EX*hslO* (TK3474). The number of particles between each spot ranged from 2 × 10^2^ to 2 × 10^6^ cells and varied 10-fold at each dilution. A combination of the photos is shown from left to right: without vitamin C at 30 °C; no vitamin C, 10 μM, 100 μM, and 1 mM vitamin C at 42 °C.

**Figure 2 ijms-23-12786-f002:**
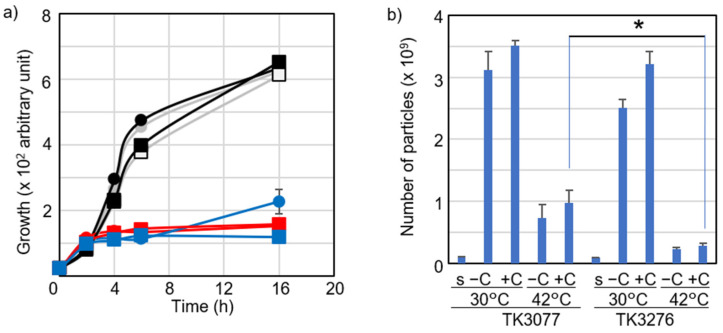
Vitamin C effect on *recAts polA* cell growth in permissive and restrictive conditions. (**a**) Optical growth inhibition at a restrictive temperature in shift-up experiments. The growth curves for the shift-up experiment at the logarithmic phase are shown. The changes in turbidity (O.D._600_ arbitrary unit) over time can be seen, starting when the culture medium is divided. TK3077 and TK3276 cells are indicated using circles and squares, respectively. The progression of cell growth is indicated by lines colored gray (without vitamin C) and black (with 100 μM vitamin C) for treatments at 30 °C. Lines are colored red (without vitamin C) and blue (with 100 μM vitamin C) for treatments at 42 °C. Each point is shown as the average and standard error of the mean (SEM) (n ≥ 3). (**b**) Cell proliferation inhibition at a restrictive temperature in shift-up experiments. Cells cultured for 16 h were measured (particles/mL) using flow cytometry. Treatment combinations are shown from left to right: seed culture (s), cell culture without vitamin C (−C), or cell culture with 100 mM vitamin C (+C) at 30 °C and 42 °C for TK3077 and TK3276 cells. Each bar represents the average and standard error of the mean (SEM) (n ≥ 3). * *p* < 0.05, according to Welch’s *t*-test (n = 6).

**Figure 3 ijms-23-12786-f003:**
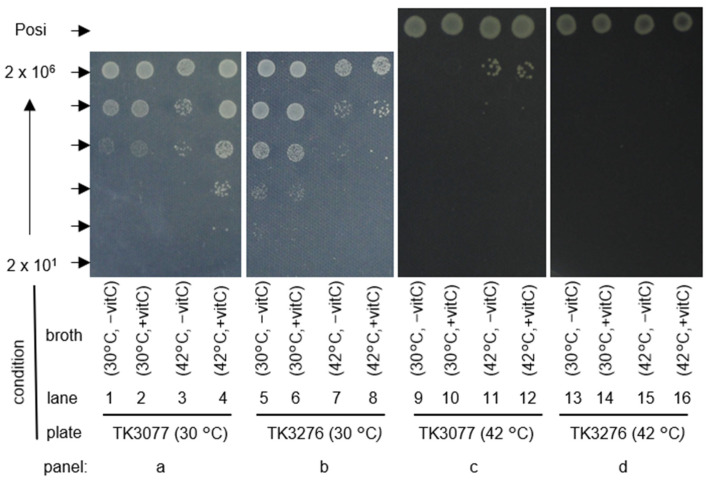
The effects of pretreatment with vitamin C on colony formation of *recAts polA* cell culture. Cells were spotted with the indicated number of particles on M9GCAA plates. Spots are shown from left to right: liquid cell culture without and with vitamin C at 30 °C and 42 °C. The number of particles between each spot (indicated with arrowheads) ranged from 2 × 10^1^ to 2 × 10^6^ cells and varied 10-fold at each dilution. A combination of the panels is shown from left to right, pretreatment (broth) at either 30 °C or 42 °C. Plate cultures (plates) were incubated at either 30 °C or 42 °C. Posi indicates 2 × 10^3^ cells of AQ10459 for spots on plates at 42 °C.

**Figure 4 ijms-23-12786-f004:**
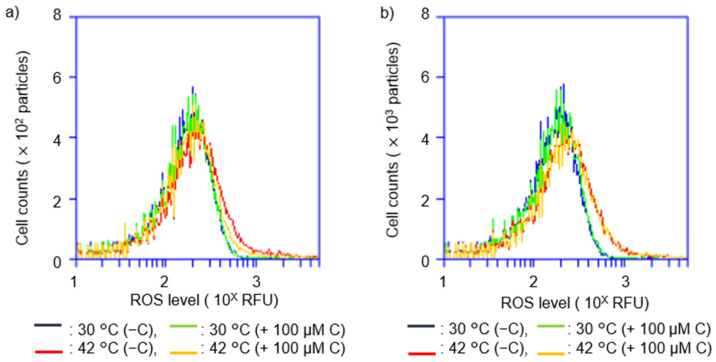
Comparison of reactive oxygen species (ROS) accumulation in restrictive and permissive conditions for *recAts polA* and *recAts polA ΔhslO* cells. (**a**) Comparison of ROS accumulation in restrictive and permissive conditions in *recAts polA* cells in a shift-up experiment. In the early logarithmic growth phase (O.D._600_ = 0.1), the TK3077 (*recAts polA*) culture was divided into three portions, and each was incubated at 30 °C and 42 °C either with or without vitamin C. The ROS levels in the TK3077 (*recAts polA*) cells were determined using CellROX Deep Red staining. ROS levels are compared at the time of division and 16 h after cultivation at 30 °C and 42 °C. Histograms show 30 °C without vitamin C (blue), 30 °C with 100 μM vitamin C (green), 42 °C without vitamin C (red), and 42 °C with 100 μM vitamin C (orange). A total of 20,000 particles are shown. (**b**) Comparison of ROS accumulation in restrictive and permissive conditions in *recAts polA ΔhslO* cells in a shift-up experiment. ROS levels of TK3276 (*recAts polA ΔhslO*) culture were determined. Histograms are labeled as in (**a**).

**Figure 5 ijms-23-12786-f005:**
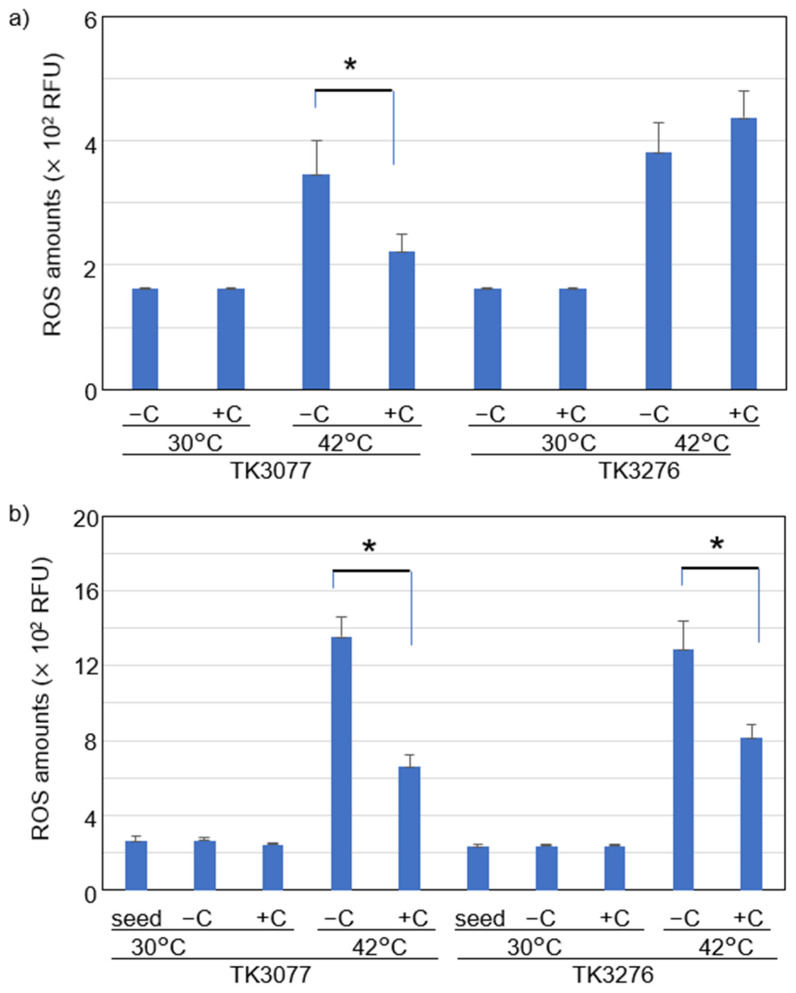
Vitamin C effect on mean reactive oxygen species (ROS) levels of *recAts polA* cells and *recAts polA ΔhslO* cells in permissive and restrictive conditions. (**a**) Mean ROS level suppression at a restrictive temperature in the shift-up experiments. In the early logarithmic growth phase (O.D._600_ = 0.1), the TK3077 (*recAts polA*) and TK3276 (*recAts polA ΔhslO*) cultures were divided into four portions, and each was incubated either with or without vitamin C at 30 °C and 42 °C. The mean ROS levels in the TK3077 and TK3276 cells were determined using CellROX Deep Red staining. The treatment combinations for each bar are shown from left to right: cell culture without vitamin C (−C) or with 100 μM vitamin C (+C) at both 30 °C and 42 °C for TK3077 and TK3276 cells. Each bar represents the average and standard error of the mean (SEM) (n ≥ 7). * *p* < 0.05, according to Welch’s *t*-test (n = 7). (**b**) Mean ROS level suppression at a restrictive temperature in inoculation experiments. TK3077 (*recAts polA*) and TK3276 (*recAts polA ΔhslO*) cells were inoculated with 1/100 of the overnight seed culture and incubated either with or without 100 μM vitamin C at 30 °C or 42 °C for 16 h. The cell ROS levels were determined, and the ROS levels of the seed and cultured cells at 30 °C or 42 °C for 16 h were compared. Treatment combinations for each bar are shown from left to right: seed culture, cell culture without vitamin C (−C), or cell culture with 100 μM vitamin C (+C) at both 30 °C and 42 °C for TK3077 and TK3276 cells. Each bar represents the average and standard error of the mean (SEM) (n ≥ 7). * *p* < 0.05, according to Welch’s *t*-test (n = 7).

**Figure 6 ijms-23-12786-f006:**
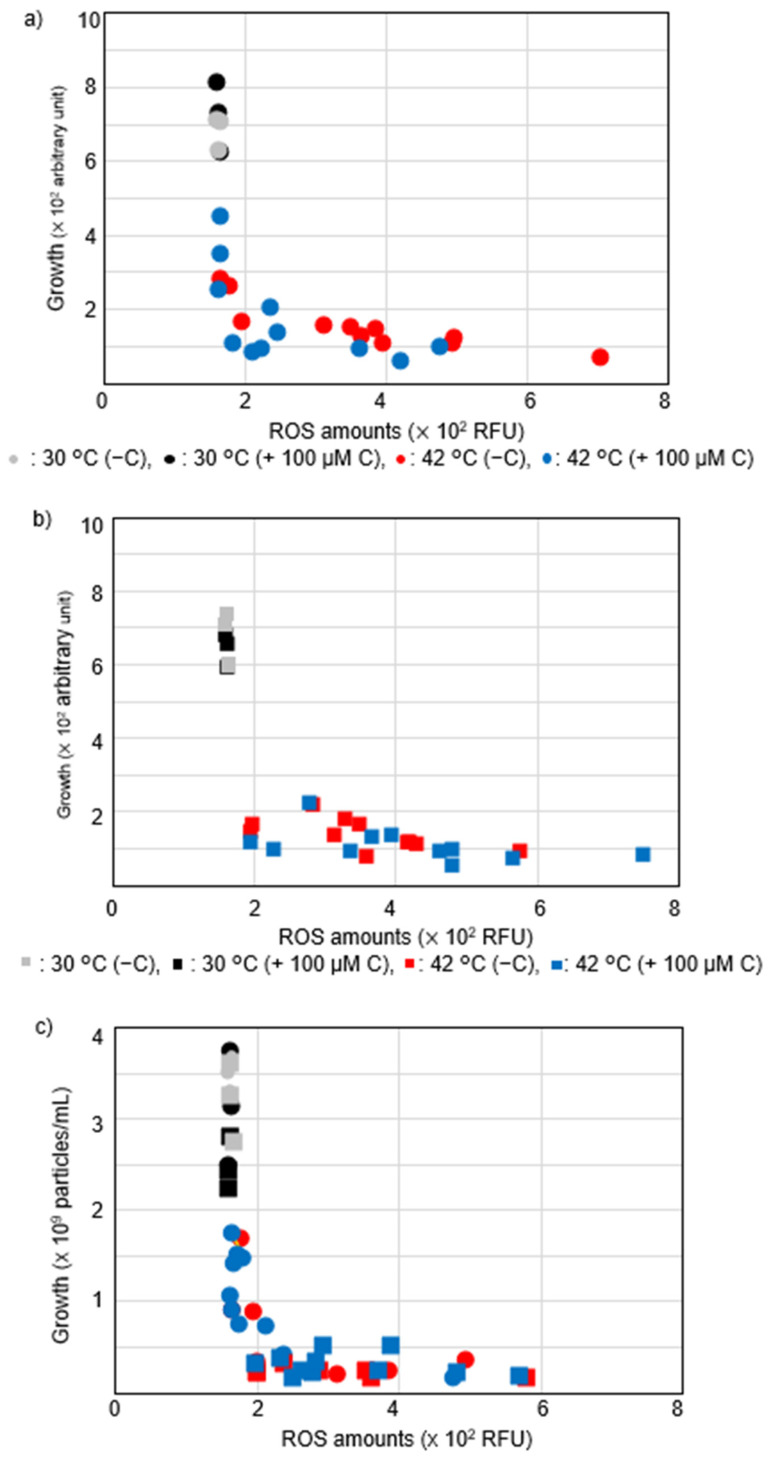
Scatterplots of how vitamin C affects mean ROS levels and cell growth in *recAts polA* cells and *recAts polA ΔhslO* cells in permissive and restrictive conditions. (**a**) Effect of vitamin C on a scatterplot with mean ROS levels and cell growth in *recAts polA* cells (TK3077) in permissive and restrictive conditions. In the early logarithmic growth phase (O.D._600_ = 0.1), the TK3077 (*recAts polA*) culture was observed. The measurements were plotted for ROS levels and optical growth on the x- and y-axis, respectively. Round symbols indicate measurements on cells without vitamin C (−C) or with 100 μM vitamin C (+100 μM C). (**b**) Vitamin C effect on a scatterplot for mean ROS levels and cell growth in *recAts polA ΔhslO* (TK3276) cells in permissive and restrictive conditions. In the early logarithmic growth phase (O.D._600_ = 0.1), the TK3276 (*recAts polA ΔhslO*) culture was examined. The results are represented just as in (**a**) using square symbols. (**c**) Vitamin C effect on a scatterplot with mean ROS levels and cell concentrations in permissive and restrictive conditions. This replot shows ROS levels and cell concentrations on the x- and *y*-axis, respectively. Measurement points are indicated as in (**a**,**b**).

**Figure 7 ijms-23-12786-f007:**
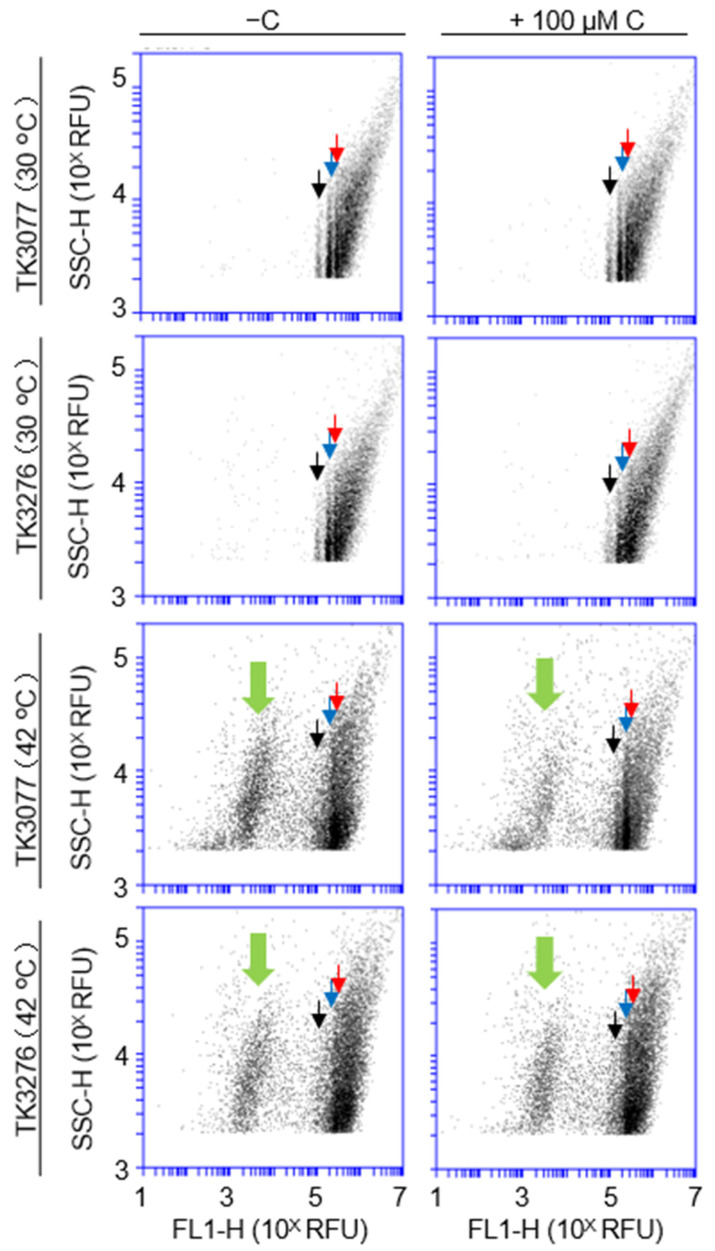
Effect of vitamin C on the ploidy of *recAts polA* cells in permissive and restrictive conditions. In the early logarithmic growth phase (O.D._600_ = 0.1), the TK3077 (*recAts polA*) and TK3276 (*recAts polA ΔhslO*) cultures were divided into four portions, and each portion was incubated at 30 °C and 42 °C either with or without vitamin C for 16 h. The DNA levels in the cells were determined using PicoGreen staining. Strains and cultivating temperatures are indicated on the left of the panels. Whether vitamin C was added (with or without 100 μM vitamin C) is indicated at the top of the panels. Arrowheads indicate ploidies as 1 (black), 2 (red), and 3 (blue). Thick green arrowheads indicate a position of anucleate cells. In total, 20,000 particles were analyzed with flow cytometry using fluorescence, with channel (FL-1H) on the *x*-axis and signal/side scatter signal (SSC) SSC-H on the *y*-axis.

**Figure 8 ijms-23-12786-f008:**
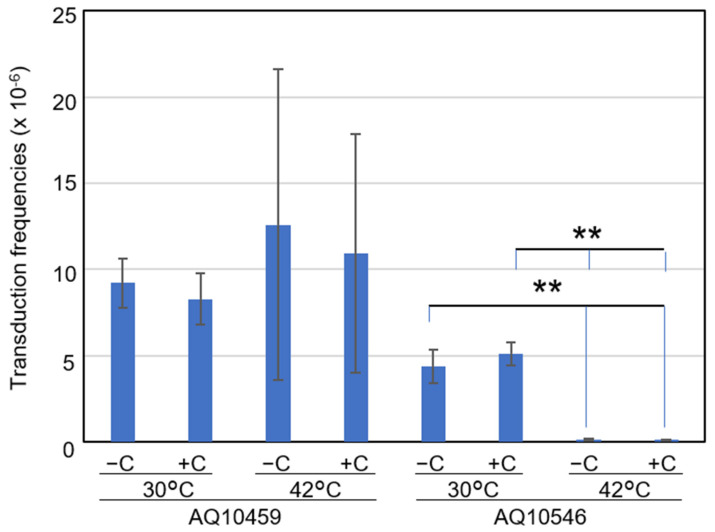
Effects of vitamin C on transduction frequencies in both permissive and restrictive temperatures. First, 10^9^ cells of either AQ10459 or AQ10546 cells were transduced with *malB*::Tn*9*. The cells with Cm resistance (Cm^r^) were subsequently selected. Further, Cm^r^ colonies were confirmed via the phenotype of maltose assimilation on MacConkey agar plates supplemented with 1% maltose. The transduction efficiencies were calculated as the ratio of Mal^−^ cells to the number of particles. The strains are AQ10459 (*recA*^+^) and AQ10546 (*recAts*). Each bar represents the average and standard error of the mean (SEM) (n ≥ 3); without (−C) or with vitamin C (+C) at 30 °C and 42 °C. ** *p* < 0.01, according to Welch’s *t*-test (n = 5).

## Data Availability

The Appendix A of this study are available from the corresponding author upon reasonable request.

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
