# Peer review of "Vitamin C Maintenance against Cell Growth Arrest and Reactive Oxygen Species Accumulation in the Presence of Redox Molecular Chaperone hslO Gene"

_ijms, 2022, doi:10.3390/ijms232112786_

Round 1
Reviewer 1 Report
This study investigates the role of vitamin C on Escherichia coli recAts polA cells growth under permissible (30 °C) or restricted temperature (42 °C) in the presence or absence of hslO gene. The author found vitamin C protect colony formation ability of Escherichia coli recAts polA cells under restricted temperature, reduce intracellular ROS levels and influence chromosome status. However, the work isn’t presented in a methodical and logical fashion. The author directly detected the effect of vitamin C on Escherichia coli recAts polA cells colonies formation in the presence or absence of hslO gene with no explanation of the relationship between vitamin C and hslO gene, which lacks rationality, thus the real contribution and significance of the research is unclear.
For results, some data were not statistically analyzed such as figure 2 and figure 8 which make the conclusions unconvincing. Additionally, the author draw conclusion with no data presented in paper, for example, the author concluded vitamin C can decrease dead cell population however there was no corresponding data in the paper. In brief, the rationality of paper and the reliability of the data are insufficient.
Issues:
1. The author concluded that vitamin C affected intracellular ROS levels by synchronizing ROS levels with growth and vitamin C influenced a converge of chromosomal DNA, however the real relationship between vitamin C's effect on cell growth and its effect on intracellular ROS and chromosomal DNA is unclear. Elucidating the relationship between vitamin C’s effect on cells growth, ROS and chromosomal DNA is necessary to understand the mechanism of vitamin C.
2. The author detected whether vitamin C reduce intracellular ROS in a dose-dependent manner using 0, 1, 10 and 100 μM of vitamin C. According to the data in figure 1, only concentration higher than 100 μM of vitamin C promoted cells to form colonies, thus lower concentration of vitamin C is insufficient to demonstrate that vitamin C affect intracellular ROS in a dose-independent manner.
3. Most of data were not clearly marked (figure 2, figure 4, figure 6 and figure 8).
4. The abstract contains lots of similar sentence compared with author’s previously published paper.
5. The conclusion in abstract that “vitamin C suppressed colony formation in recAts polA cells growth under restricted temperature” is inconsistent with results.
6. In line 123, the author attempted to investigate the effect of vitamin C on ROS accumulation in liquid media, however the actual experiment was detecting the role of vitamin C on recAts polA cell growth.
Author Response
IJMS Revewer1
Comment and suggestions for author Comment: This study investigates the role of vitamin C on Escherichia coli recAts polA cells growth under permissible (30 °C) or restricted temperature (42 °C) in the presence or absence of hslO gene. The author found vitamin C protect colony formation ability of Escherichia coli recAts polA cells under restricted temperature, reduce intracellular ROS levels and influence chromosome status. However, the work isn’t presented in a methodical and logical fashion. The author directly detected the effect of vitamin C on Escherichia coli recAts polA cells colonies formation in the presence or absence of hslO gene with no explanation of the relationship between vitamin C and hslO gene, which lacks rationality, thus the real contribution and significance of the research is unclear.
Response: Thank you for your honest feedback and valuable assessments. We have revised the manuscript according to the suggestions and comments. Our revisions have been highlighted in yellow. As in the comment and Issue 1, we think there are certainly some shortcomings, and we have attempted to revise the manuscript to better showcase our logical reasoning to a degree that can be allowed by our English writing skills. Regarding the last portion of your suggestion, we have revised the manuscript by adding some interpretations based on our present findings that will hopefully clarify our reasoning and the findings’ novelty. Kindly return to the manuscript of this submission to ensure that we have corrected it to your satisfaction and within the scope of our present results.
For results, some data were not statistically analyzed such as figure 2 and figure 8 which make the conclusions unconvincing. Additionally, the author draw conclusion with no data presented in paper, for example, the author concluded vitamin C can decrease dead cell population however there was no corresponding data in the paper. In brief, the rationality of paper and the reliability of the data are insufficient.
Response: Thank you for pointing this out. We have appended the results of Welch's t-tests with respect to Figures 2 (page 4) and Figure 8 (page 12). We have tried to provide reasonable explanations based on the data currently available in the figures. As for our claims on dead cells, we have not clarified and confirmed the dead cells by separating them from the other populations. We deduced that the dead cell population decreased by the process of elimination. We have nonetheless re-organized our arguments for the areas that you have pointed out, and if we cannot judge our arguments to be self-evident, we will correct the text by deleting them. Thank you for pointing this out. The following is a sample of the changes made to this section: As regard to Figure 2, line134-141 “Nonetheless, a significant difference in cell particles was observed between TK3077 (in the presence of hsLO) and TK3276 (in the absence of hslO) at 42 °C with the addition of 100 μM vitamin C (Welch’s t-test, (p<0.05). It indicated that hslO is necessary for vitamin C to stimulate a slight increment of cell growth and particles. Colony formations were remarkably restored in TK3077 cells with vitamin C at 42°C (Figure 1), in spite of only small increase in those of cell particles. Thus, it was necessary to examine those cell particles whether those cell particles were membrane damaged or not. Therefore, we assayed dead cells with compromised membrane integrity by propidium iodide (PI) staining. Membrane-damaged dead cells were not detected at 30 °C (Figure S1ab), but were detected at 42 °C, as indicated by blue arrowheads (Figure S1cd). The membrane-damaged cell population (blue arrowhead) only slightly shrank following vitamin C addition. Meanwhile, a minor cell population staining poorly for both PI and SYTO9 (red arrowhead) had also shrunk. This suggested that vitamin C addition did not remarkably reduce the number of membrane-damaged cells. It is unclear whether this small change was enough to explain the observed increase in colony formation in Figure 1. Furthermore, this does not take into consideration any possible decrease in the number of anucleate cells. Therefore, we considered whether qualitative rather than quantitative changes in chromosomal DNA might be involved in the cell viability decrement at restricted temperature in recAts polA cells. “(Lines 134 – 152, pages 3-4)
Issues: 1. The author concluded that vitamin C affected intracellular ROS levels by synchronizing ROS levels with growth and vitamin C influenced a converge of chromosomal DNA, however the real relationship between vitamin C's effect on cell growth and its effect on intracellular ROS and chromosomal DNA is unclear. Elucidating the relationship between vitamin C’s effect on cells growth, ROS and chromosomal DNA is necessary to understand the mechanism of vitamin C.
Response: Thank you for your feedback. What we are currently able to confirm is the synchrony between ROS levels and growth or cellular maintenance. It is unclear whether growth is maintained for any reason, as you noted. As for chromosomal convergence, recAts polA lethality has been attributed to chromosome degradation in 1995 (Cao and Kogoma, 1995). We have reported in a previous paper that significant chromosome breakdown could not be observed (Kaidow et al, 2022). However, it also indicated that we no longer know what causes the growth failure of the recAts polA cells. In the previous paper, we reported that ROS accumulation synchronized with growth failure. In this paper, we instead focused on the process leading to ROS accumulation and investigated what changed in the absence or presence of the hslO gene. On the other hand, because recAts polA cells were expected to have chromosomal effects, this manuscript attempted to characterize the damaged chromosomes. As a result, we have confirmed that recAts polA cells could grow in the presence of both the hslO gene and vitamin C under restrictive conditions. Conversely, we reported that recAts polA cells failed a converge of cell population along with their DNA content at the restrictive temperature, but this phenomenon was restored in the presence of hslO and vitamin C. As you pointed out, we have not been able to determine what vitamin C does to the chromosome convergence mechanism. Conversely, we have not been able to answer whether the failure of chromosome converge is enough to cause the growth failure directly. It is simply one of the interesting phenotypes that synchronize with the vitamin C-induced growth restoration. However, it is true that slowly growing cells are possessed 1 or 2 ploidies, as we mentioned a converge of cell population along their DNA content, and those converge, in other words completion of replication, must be required for those slowly growing cells. It is most likely that oxidation sensitive character of DNA polymerase will be involved. As you pointed out, the paper does not answer how vitamin C works in these phenotypes. However, this manuscript does indicate that hslO is the key. Regarding the above, this paper focused on the factors relating to the phenotype and clarifies what should be the focus in the future. The "relationship between the effects of vitamin C on cell growth, reactive oxygen species, and chromosomal DNA" that you mentioned is one of our final goals. The elucidation of the mechanism is still incomplete. We would appreciate it if you could take these points into consideration. Thank you for pointing this out.
2. The author detected whether vitamin C reduce intracellular ROS in a dose-dependent manner using 0, 1, 10 and 100 μM of vitamin C. According to the data in figure 1, only concentration higher than 100 μM of vitamin C promoted cells to form colonies, thus lower concentration of vitamin C is insufficient to demonstrate that vitamin C affect intracellular ROS in a dose-independent manner.
Response: Thank you for pointing this out. You have mentioned that the phenomenon was observed at 100 µM vitamin C. If we have understood correctly, you also would like us to consider other concentrations of vitamin C beyond what had been presented in our figure. We have also tried to confirm these effects at the 30 μM vitaminC, where suppression was inconsistent, so they are not shown here. On the other hand, we had also attempted to test a higher concentration, 300 μM and 1 mM vitamin C. The result was similar to that of 100 µM vitamin C, but the result was also inconsistent. We think these unstable results occurred due to a hormesis-like effect as described in Fig. S2. For this reason, this paper uses 100 μM vitamin C in later experiments. It is certainly possible that ascorbic acid radicals could have also affected the system, due to a reaction of vitamin C with reactive oxygen species. Since this is a report of a phenomenon, we have not examined the causes.
3. Most of data were not clearly marked (figure 2, figure 4, figure 6 and figure 8).
Response: Thank you for letting us know of this issue. We have rebuilt the figures with larger fonts so that the data can be easily interpreted (see Figure 2 on page 4, Figure 4 on page 6, Figure 6 on page 10, and Figure 8 on page 13).
4. The abstract contains lots of similar sentence compared with author’s previously published paper.
Response: Thank you for pointing this out. I rewrite abstract as your suggestion with correction indicated in yellow as follows. “Subsequently, we investigated the role of hslO in the cell growth failure of recAts polA cells. The effects of vitamin C were observed in hsl+O cells; simultaneously, cells converging along several ploidies likely through a completion of replication, with the addition of vitamin C at restricted temperatures. These results suggest that HslO could manage oxidative stress to an acceptable level, satisfying prerequisite for cell division and rescues cell growth. Overall, ROS may regulate several processes, from damage response to cell division. Our results might be clue for unsolved regulatory interplay of cellular processes.”
5. The conclusion in abstract that “vitamin C suppressed colony formation in recAts polA cells growth under restricted temperature” is inconsistent with results.
Response: Thank you very much for pointing out this error. It seems we have mistranslated this sentence from our original Japanese wording. We have corrected it as follows: " In this study, we used a reducing agent, vitamin C, to confirm whether cell growth could be improved. Vitamin C reduced ROS levels and rescued colony formation in recAts polA cells under restrictive temperatures. " (Lines 14 – 16, page 1)
6. In line 123, the author attempted to investigate the effect of vitamin C on ROS accumulation in liquid media, however the actual experiment was detecting the role of vitamin C on recAts polA cell growth.
Response: Thank you for pointing this out. This was an error on our part, since we were thinking about the connections to ROS based on the results in Figure 2, while we were writing this sentence. I corrected it as follows: " To directly assess the effect of vitamin C on cell growth, we used liquid media, in which cell growth could be easily measured.” (Line 126 – 127, page 3) Closing statement: My coauthors and I thank the reviewers and editors for the time they took to provide us with such valuable feedback. We look forward to hearing from you and would make any necessary changes, if required.
Reviewer 2 Report
In the previous paper, the authors showed that growth inhibition of the Escherichia coli recA polA mutant cells under restricted conditions was accompanied with accumulation of reactive oxygen species (ROS). In this manuscript, they showed that accumulation of ROS was partly suppressed by the addition of vitamin C only under the hslO+ genetic background. In addition, chromosome ploidy analysis showed that HslO supports the completion of chromosome replication in the presence of vitamin C.
Major comments
1) In the previous paper, the authors showed that the recA polA lethality was suppressed by the lexA51 mutation, which induces the expression of the hslO gene. In this manuscript, they showed that the colony forming ability of the recA polA mutant was recovered, although partially, with the supplementation of vitamin C only under the hslO+ genetic background. It is possible that the vitamin C supplementation induces the expression of the hslO. Expression levels of the hslO gene should be checked with or without vitamin C supplementation.
2) The authors argued that the abnormality of chromosome ploidy of the recA polA mutant was suppressed by the addition of vitamin C based on the results of flow cytometry analysis. However, it is difficult, at least for me, to evaluate the results of Figure 7. Please indicate in the figure which signals represent the ploidies 1, 2 and 3.
The legend of Figure 7 does not match the figure.
Minor comments
There are so many mistakes in the manuscripts
1) Figure 1.
10 mM and 100 mM should be read as 10 mM and 100 mM, respectively.
2) Line 144. “This observation is in agreement with our previous observation.”
The reference should be cited.
3) Figure 2A (line 127) and Figure 2B (line 129) should be read as Figure 2a and Figure 2b, respectively. There are so many similar mistakes.
4) Line 186. (upper right).
Which one is upper right?
5) Gene name and species name should be italicized in References.
Author Response
IJMS Revewer2
Comment and suggestions for author
In the previous paper, the authors showed that growth inhibition of the Escherichia coli recA polA mutant cells under restricted conditions was accompanied with accumulation of reactive oxygen species (ROS). In this manuscript, they showed that accumulation of ROS was partly suppressed by the addition of vitamin C only under the hslO+ genetic background. In addition, chromosome ploidy analysis showed that HslO supports the completion of chromosome replication in the presence of vitamin C.
Response: Thank you for your reading this manuscript.
Major comments
1) In the previous paper, the authors showed that the recA polA lethality was suppressed by the lexA51 mutation, which induces the expression of the hslO gene. In this manuscript, they showed that the colony forming ability of the recA polA mutant was recovered, although partially, with the supplementation of vitamin C only under the hslO+ genetic background. It is possible that the vitamin C supplementation induces the expression of the hslO. Expression levels of the hslO gene should be checked with or without vitamin C supplementation.
Response: Thank you for mentioning this. We have received similar suggestions from the other reviewers in our previously published reports. The current paper aimed to confirm the phenotype at the cellular level, and our analysis was based on whether the hslO gene was present or not. On the other hand, I agree that it is better to confirm the expression as you pointed out. Although it is difficult to detect HslO using antibodies due to the protein’s low expression level, I think it is possible to confirm hslO expression by RT-PCR. If RT-PCR is a prerequisite for the next round, we will require more time to prepare this experiment. We would like to purchase transcriptase and RNA preparation kits and plan these additional experiments during this time. I am deeply grateful for your suggestions, as they would certainly enrich our analysis.
2) The authors argued that the abnormality of chromosome ploidy of the recA polA mutant was suppressed by the addition of vitamin C based on the results of flow cytometry analysis. However, it is difficult, at least for me, to evaluate the results of Figure 7. Please indicate in the figure which signals represent the ploidies 1, 2 and 3.
Response: Thank you for your feedback. We have modified Figure 7 (page 12) to indicate and distinguish the bands associated with ploidies with differently colored arrow heads.
3) The legend of Figure 7 does not match the figure. Response: Thank you for this observation. We have added the correct legend for Figure 7 (lines 404 – 412, page13). Minor comments There are so many mistakes in the manuscripts 1) Figure 1. 10 mM and 100 mM should be read as XXXXX, respectively.
Response: Thank you for bringing this to our attention. The figure had been incorrectly converted when we changed the font for submission. We have corrected the concentrations to µM (Figure 1, page 3). 2) Line 144. “This observation is in agreement with our previous observation.” The reference should be cited.
Response: Thank you for bringing this to our attention. I had added the aforementioned studies. The reference should be as follows: Reference 2: Kaidow A.; Ishii N.; Suzuki S.; Shiina T.; Kasahara H. Reactive oxygen species accumulation is synchronized with growth inhibition of temperature-sensitive recAts polA Escherichia coli. Arch. Microbiol. 2022, 204, 396 [DOI:10.1007/s00203-022-02957-z]. However, during corrections, corresponding sentence is now as follows. “as indicated by blue arrowheads (Figure S1cd). The membrane-damaged cell population (blue arrowhead) only slightly shrank following vitamin C addition. Meanwhile, a minor cell population staining poorly for both PI and SYTO9 (red arrowhead) had also shrunk. This suggested that vitamin C addition did not remarkably reduce the number of membrane-damaged cells. It is unclear whether this small change was enough to explain the observed increase in colony formation in Figure 1. Furthermore, this does not take into consideration any possible decrease in the number of anucleate cells. Therefore, we considered whether qualitative rather than quantitative changes in chromosomal DNA might be involved in the cell viability decrement at restricted temperature in recAts polA cells.”(lines143-152, page 4)
Thank you for your pointed out. I am sorry that corresponding sentence is missing in corrected manuscript now, so I can’t add reference.
Figure 2A (line 127) and Figure 2B (line 129) should be read as Figure 2a and Figure 2b, respectively. There are so many similar mistakes.
Response: Thank you for pointing out these errors. We apologize if this has caused any inconvenience in the review of our manuscript. Accordingly, we have rectified these errors: “As shown in Figure 2a, the recAts polA cells (TK3077) at 42°C showed slightly improved growth at 16 h with the addition of 100 μM vitamin C (blue), compared to cells without vitamin C (red). The number of bacterial cells at 16 h is also shown in Figure 2b.” (Lines 130 –132, page 3) We have also taken care to check for similar errors throughout the manuscript.
4) Line 186. (upper right). Which one is upper right?
Response: Thank you for letting us know about this. We forgot to correct it when we rebuilt the figure from a two-column figure to fit the IJMS journal. It has been changed as follows: “These cells showed no substantial difference in colony-forming abilities between those with or without vitamin C supplementation at both 30 °C and 42 °C treatment (lanes 5 to 8 in the panel b)” (Lines 198, page 5)
5) Gene name and species name should be italicized in References.
Response: Thank you for your observation. I corrected the gene name accordingly as italics.
Closing statement: My coauthors and I thank the reviewers and editors for the time they took to provide us with such valuable feedback. We look forward to hearing from you and would make any necessary changes, if required.
Round 2
Reviewer 1 Report
The authors partly addressed my comments, thanks.
1. The relationship between vitamin C's effect on cell growth, intracellular ROS and chromosomal DNA need to be added in discussion section.
2. For the results about whether vitamin C reduce intracellular ROS in a dose-dependent manner, the author mentioned that 100 μM vitamin C has effect, 300 μM and 1 mM vitamin C has similar effect with 100 μM vitamin C however the result was inconsistent. What does “inconsistent result” mean? Data need to be provided and illustrated them in manuscript.
3. The figure S2 presented in paper is same with the figure S3.
4. In line 140-142, the author mentioned that vitamin C slightly shrank the membrane-damaged cell population, however there is no data supported this, data need to be provided.
Author Response
Dear Editor and reviewers,
Thank you for your warm response.
I have provided point-by-point responses to your comments/suggestion below. Please check accordingly.
1. The relationship between vitamin C's effect on cell growth, intracellular ROS and chromosomal DNA need to be added in discussion section.
Response: Thank you for pointing this out. We have added a section to the Discussion as follows (pink highlighted): “We have reported in a previous paper that significant chromosome breakdown could not be observed [2]. However, we were not able to identify what causes growth failure of the recAts polA cells. Therefore, we instead focused on the process leading to ROS accumulation and also characterized the damaged chromosomes. As a result, recAts polA cells could grow in the presence of both the hslO gene and vitamin C under restrictive conditions. Conversely, we found that recAts polA cells failed a converge of cell population along with their DNA content at the restrictive temperature; however, this phenomenon was restored in the presence of hslO and vitamin C. It is not yet known whether the growth failure and the convergence of cell population about DNA content are related each other. It is interesting that slowly growing cells possess one or two ploidies, indicating that completion of replication was always observed for those slowly growing cells before cell proliferation via cell division. Meanwhile, oxidation of DNA polymerase is likely leading their inactivation. Therefore, hslO and vitamin C can function together to maintain cellular oxidative conditions in allowable levels, supporting cellular metabolism and the completion of DNA replication required for the convergence of cell population along their DNA contents, thus enabling cell division and cell proliferation.”
2. For the results about whether vitamin C reduce intracellular ROS in a dose-dependent manner, the author mentioned that 100 μM vitamin C has effect, 300 μM and 1 mM vitamin C has similar effect with 100 μM vitamin C however the result was inconsistent. What does “inconsistent result” mean? Data need to be provided and illustrated them in manuscript.
Response: Thank you for pointing this out. It means that the results seem unstable. Therefore, we think "inconsistent" is a better word for this purpose. In the revised manuscript, we have described the corresponding part as follows (pink highlighted): “Next, we analyzed the dose-dependent response in mean ROS levels towards vitamin C. We observed poor dose response; however, the effect of vitamin C was observed at 100 μM in TK3077 (Figure S2). This poor dose response and unstable results at both 300 and 1000 μM might be due to a hormesis effect [24], in which a response was not always according to their dosages.”
3. The figure S2 presented in paper is same with the figure S3.
Response: Thank you for pointing this out. We have replaced Figure S2 and added 300 and 1000 μM vitamin C as per your valuable suggestion.
4. In line 140-142, the author mentioned that vitamin C slightly shrank the membrane-damaged cell population, however there is no data supported this, data need to be provided.
Response: Thank you for pointing this out. In Figure S1, we wanted to show that membrane-damaged population (blue arrows) was remarkably recognized at a restricted temperature; however, this membrane-damaged population was not a major cell population in those cells. We mentioned the presence of only a slight decrease to this damaged population after the supplementation of vitamin C; however, we wanted to show that a decrement of damaged population was not sufficient in explaining an improvement of the viability observed in Figure 1. Thus, we would like to apologize for this discrepancy. We have changed the corresponding sentence as follows: "However, contrary to our expectation, the membrane-damaged cell population (blue arrowheads) did not show remarkable decrease following vitamin C addition.
" Closing statement: My coauthors and I thank the reviewers and editors for the time they took to provide us with such valuable feedback. We look forward to hearing from you and would make any necessary changes, if required.
Round 3
Reviewer 1 Report
Thanks for addressing my comments. I have no further comments on experiments.